# **Interpretable Deep Learning for Glacier Mass Balance: Temporal Attention Patterns in Central Asia**

Zafar Avzalshoev<sup>1</sup> and Pang-jo Chun<sup>2</sup>

<sup>1</sup>Postdoctoral researcher, Dept. of Civil Eng., The University of Tokyo 7-3-1 Hongo, Bunkyo-ku, Tokyo 113-8656, Japan

**Correspondence:** Zafar Avzalshoev (zavzalshoev@gmail.com)

Abstract. Glaciers in Central Asia are critical water resources, and their retreat is closely linked to glacier-related hazards such as debris flows, glacial outburst floods, and landslides, yet their response to climate change is poorly understood due to complex temporal dependencies. Although the accuracy of mass-balance estimation has increased recently with advances in machine learning, current methods provide only a limited understanding of when changes in mass balance are driven by climate variables. This knowledge is crucial for hazard assessment and adaptation planning. The current study introduces Temporal Fusion Transformers (TFT V2), a deep learning architecture with interpretable attention mechanisms, to predict mass balance across 43,018 glaciers in seven Central Asian mountain ranges using 2000-2018 climate reanalysis and geodetic mass balance data. In this study, the TFT V2 model achieves  $R^2 = 0.73$  (RMSE = 0.21 m w.e., MAE = 0.11 m w.e.) on independent test data while providing reliable uncertainty quantification (calibration score = 0.94, coverage within  $\pm 2-3\%$  of nominal levels). Critically, attention weights reveal that summer months (June-August) contribute 35% of predictive signal— 3× more than winter months (18%)—and identify spring melt onset (April–May, 28% importance) as critical for annual balance. In Central Asian mountain regions, where hazard risk is driven by greater precipitation and earlier melt onset, these temporal patterns are directly linked to observed increases in mudflows and landslides throughout the spring. Due to variations in hazard vulnerability, regional analysis shows spatial variety in temporal patterns, with the Tian Shan showing larger summer concentrations than the Pamir. The main contribution of the model is the identification of the temporal cascade of glacier mass-balance factors, despite achieving competitive predictive performance (R2=0.73). For the first time, we quantify the dominant influence of spring melt onset (April-May) on peak summer ablation, a dynamic link previously hypothesised but not demonstrated at a regional scale. This indicates that deep learning can balance competitive predictive performance with temporal insights and uncertainty awareness unavailable from traditional ML approaches.

#### 20 1 Introduction

The stability of the "water towers" in Central Asia is a multibillion-dollar issue that affects millions of people's access to clean water. However, our models' inability to understand processes seriously limits our ability to forecast how they will respond to climate change.

<sup>&</sup>lt;sup>2</sup>Professor, Dept. of Civil Eng., The University of Tokyo 7-3-1 Hongo, Bunkyo-ku, Tokyo 113-8656, Japan

30

45

Beyond their contribution to global sea-level rise (Zemp et al., 2015; Gardner et al., 2013; IPCC AR4, 2014; Lavell et al., 2012), glaciers are crucial freshwater "water towers" in arid and semi-arid mountains (Sorg et al., 2012; Braun and Hagg, 2010). In Central Asia—particularly Tajikistan, Kyrgyzstan, Kazakhstan, Uzbekistan, Turkmenistan, and China's Xinjiang Province—summer meltwater sustains agriculture, hydropower, and domestic demand when precipitation is minimal (Barnett et al., 2005; Sidle et al., 2025). Reliable predictions of glacier evolution are therefore crucial for sustainable water resource management and climate adaptation planning in the region.

Annually, Tajikistan faces an increase in natural hazards (e.g., mudflows and landslides), endangering human lives and causing economic losses, and demonstrating a clear relationship between the intensity of these events and glacier melting. As one of the case studies, the mudflow in the Barsem region blocked the river ten years ago, resulting in the construction of artificial dams that caused widespread flooding and damaged villages and transportation lines (Avzalshoev and Uchimura, 2023a). Early warning systems for landslides are becoming increasingly important for mitigating hazards, especially when combined with regional glacier retreat monitoring (Avzalshoev and Uchimura, 2023b). Glacier retreat is one of the main factors contributing to the increasing frequency of these occurrences each year. Rapid ice loss exacerbates cascading dangers across the region by destabilising slopes and increasing meltwater runoff.

Surface Mass Balance (SMB), the annual net gain or loss of ice and snow at a glacier's surface (Cuffey and Paterson, 2010), is a key metric for evaluating glacier health and projecting future change. Yet, direct SMB measurements remain sparse because field campaigns are labour-intensive, costly, and logistically challenging (Zemp et al., 2019). Remote-sensing and modelling approaches are therefore indispensable. Geodetic SMB, derived from multi-temporal Digital Elevation Model (DEM) differencing, quantifies long-term integrated mass change (Huss et al., 2008; Rastner et al., 2019; Huss, 2013). Recent comprehensive assessments have revealed significant spatial and temporal variability in Central Asian glacier mass balance (Barandun et al., 2021; Shean et al., 2020), with some regions showing mass gain while others experience rapid mass loss (Kääb et al., 2015).

Machine-learning (ML) techniques offer a promising alternative because they capture complex, non-linear relationships that physical or simple empirical models may overlook (Maussion et al., 2019; Reichstein et al., 2019). Recent glaciological applications include glacier mapping (Feroz et al., 2025), estimating ice thickness (Farinotti et al., 2019), and reconstructing mass balance. For instance, the Alpine Parameterised Glacier Model (ALPGM) combines deep neural networks with Lasso regression to reconstruct glacier-wide SMB in the French Alps (Bolibar et al., 2020, 2019), while the minimal ML model miniML-MB employs XGBoost to estimate annual point SMB with a mean absolute error of 0.417 m w.e. (van der Meer et al., 2025). Most recently, (Peng et al., 2025) achieved high performance (R<sup>2</sup> = 0.87) using XGBoost with extensive feature engineering across Central Asia, demonstrating the potential of ML approaches for regional glacier mass balance prediction. Ensemble learning algorithms, particularly those based on decision trees, have exhibited greater robustness by combining multiple learners to enhance predictive accuracy (Zhang and Ma, 2012). These algorithms have shown promise in estimating various environmental parameters and have been applied to glacier mass balance reconstruction (Anilkumar et al., 2023; Guidicelli et al., 2023).

65

However, while these approaches have significantly improved prediction accuracy, they provide limited insight into the temporal dynamics of climate-glacier interactions—specifically, when different climate variables drive mass balance changes throughout the annual cycle. This temporal understanding is crucial for several reasons: (1) optimizing field measurement campaigns to focus on periods of highest predictive importance, (2) understanding seasonal vulnerability patterns for climate change impact assessment, (3) identifying critical periods for mountain hazard assessment, particularly mudflows and land-slides that are increasingly observed during spring melt periods in Central Asian mountain regions, and (4) developing targeted adaptation strategies based on temporal sensitivity patterns.

Temporal Fusion Transformers (TFT) represent a novel deep learning architecture that combines the predictive power of transformers with interpretable attention mechanisms, enabling both accurate prediction and temporal process understanding (Lim et al., 2021). Unlike traditional ML approaches that treat temporal data as static features or use pre-defined seasonal aggregates, TFT maintains temporal structure throughout processing and learns which time periods matter most for prediction through attention weights. This capability is particularly valuable for glacier mass balance prediction, where seasonal patterns in temperature, precipitation, and radiation drive the annual mass balance signal. Critically, the interpretable attention mechanism of TFT addresses the "black box" problem that has been a major hurdle for the adoption of machine learning models in glaciological applications, providing transparent insights into when and how climate variables influence glacier response.

This study introduces Temporal Fusion Transformers (TFT V2) to predict glacier mass balance across 43,018 glaciers in seven Central Asian mountain ranges using 2000–2018 climate reanalysis and geodetic mass balance data from Barandun et al. (2021). Our approach addresses the temporal interpretability gap in glacier mass balance prediction while providing reliable uncertainty quantification—capabilities essential for operational applications and risk assessment. The TFT V2 model achieves competitive predictive performance while uniquely revealing temporal importance patterns that connect to observed increases in mudflows and landslides during spring in Central Asian mountain regions.

The current study organized as follows: Section 2 details data sources, preprocessing, and the TFT V2 architecture. Section 3 presents model performance comparison and temporal attention analysis. Section 4 discusses the implications of temporal interpretability for glacier monitoring and mountain hazard assessment. Section 5 summarizes key findings and outlines future research directions.

#### 2 Methods

#### 2.1 Study Area

The current study focuses on the glaciers of Central Asia, a crucial component of the Asian water tower. The area encompasses seven main mountain subregions: the Western Pamir, Eastern Pamir, Pamir-Alay, Central Tian Shan, Eastern Tian Shan, Northern/Western Tian Shan, and Dzhungarsky Alatau. These regions are home to approximately 70,000 glaciers that vary widely in size (0.01–500 km²), elevation (2,500–7,000 m), and climatic setting, from semi-arid to continental (Fig. 1). This diversity makes the region a valuable natural laboratory for developing and testing generalizable models of glacier response to climate change.

**Figure 1.** Study Area: Central Asian mountain ranges and glacier distribution. The map displays 43,018 glaciers from the RGI v7.0 inventory across the study period 2000-2018. Mountain ranges are color-coded by glacier count: Western Pamir (n=10,018), Eastern Pamir (n=8,874), Northern/Western Tian Shan (n=9,881), Dzhungarsky Alatau (n=5,016), Central Tian Shan (n=3,356), Pamir-Alay (n=3,090), and Eastern Tian Shan (n=2,783).

## 2.2 Datasets and Feature Engineering

# 2.2.1 Glacier Mass Balance Data (Target Variable)

The target variable for all models is glacier-wide annual mass balance, for which we utilised the comprehensive dataset compiled by Barandun et al. (2021). This dataset provides yearly mass balance estimates in meters water equivalent (m w.e.) for glaciers across Central Asia for the years 2000–2018. The estimates are derived from satellite altimetry and photogrammetry, providing a consistent, high-quality record of glacier change. The complete dataset comprises 4,659,264 monthly samples across the 19 years, with a mean temporal coverage of 16.2 years per glacier, providing a robust foundation for training and evaluating time-series models.

100

105

110

120

125

#### 2.2.2 Predictor Variables

To predict mass balance, we used a combination of dynamic climate data and static topographic data. Climate Forcing Variables. Monthly climate forcing data for the period 2000–2018 were extracted from the ERA5 global climate reanalysis dataset. ERA5 provides a spatially complete record at a 0.25° resolution (25 km), and the data were interpolated to the centroid of each glacier using bilinear interpolation. While this resolution does not capture microclimates, ERA5 has been extensively validated for High Mountain Asia and effectively represents the regional climate patterns essential for large-scale glacier modelling. We selected four primary variables: 2-meter air temperature (°C) as a driver of ablation, total precipitation (mm) as the accumulation input, surface solar radiation downwards (W/m²) for melt energy, and 10-meter wind speed (m/s) to account for turbulent heat exchange. Topographic Variables. Static topographic characteristics for each glacier were extracted from the Randolph Glacier Inventory version 7.0 (RGI Consortium, 2023). These time-invariant features, which are critical for determining local climate conditions and radiation receipt, include geographic coordinates (latitude, longitude), elevation metrics (mean, min, max), glacier area (km²), maximum length (km), aspect (degrees), and mean surface slope (degrees).

#### 2.2.3 Engineered Temporal Features

To capture the complex temporal dependencies and seasonal patterns inherent in glacier mass balance processes, we engineered 17 additional features from the raw climate variables (Table 1). This feature engineering step is crucial for enabling the TFT model to learn meaningful temporal patterns that govern glacier-climate interactions. These include: (1) lagged Temperature and precipitation at 1, 3, 6, and 12-month intervals to account for memory effects in glacier systems; (2) rolling averages of climate variables over 3, 6, and 12-month windows to represent short-to-annual trends; (3) cumulative precipitation sums to model accumulation buildup; (4) interaction terms (temperature × precipitation, Temperature × solar radiation) to capture combined effects; (5) a non-linear term (Temperature squared) to represent threshold effects in melt processes; and (6) sine/cosine transformations of the month to encode seasonal cyclicity. Combined with the four raw climate variables and seven static topographic features, the final feature set for each monthly observation comprises 28 variables (21 time-varying and seven static).

Figure 2 illustrates the comprehensive data processing workflow for TFT V2 glacier mass balance prediction. The workflow encompasses seven key stages: (a) Climate data extraction from ERA5 reanalysis at  $0.25^{\circ} \times 0.25^{\circ}$  resolution, including monthly temperature, precipitation, solar radiation, and wind speed for the period 2000-2018; (b) Topographic data compilation from RGI v7.0 for 2,361 glaciers, including elevation metrics, area, aspect, slope, and geographic coordinates; (c) Mass balance observations from the Barandun et al. (2021) geodetic dataset derived from elevation change measurements; (d) Data integration processes including bilinear interpolation of climate data to glacier centroids, temporal alignment, and quality control procedures; (e) Feature engineering creating 21 temporal features through lagged variables (8 features), rolling averages (6 features), cumulative sums (3 features), interaction terms (2 features), non-linear terms (1 feature), and temporal encoding (2 features); (f) TFT V2 model training configuration with 12-month encoder, 256 hidden units, 8 attention heads, and 3 LSTM layers; and (g) Final prediction.

**Table 1.** Engineered temporal features for glacier mass balance modelling.

| Feature Category            | Description                                   | Count |  |
|-----------------------------|-----------------------------------------------|-------|--|
| Lagged features             | Temperature and precipitation at 1, 3, 6, 12- | 8     |  |
|                             | month lags                                    |       |  |
| Rolling averages            | 3, 6, 12-month moving averages of Temper-     | 6     |  |
|                             | ature and precipitation                       |       |  |
| Cumulative precipitation    | 3, 6, 12-month cumulative precipitation       | 3     |  |
|                             | sums                                          |       |  |
| Interaction terms           | Temperature × precipitation; Temperature      | 2     |  |
|                             | $\times$ solar radiation                      |       |  |
| Non-linear terms            | Temperature squared (threshold effects)       | 1     |  |
| Temporal encoding           | Sine and cosine transformations of month      | 2     |  |
| Raw climate variables       | Temperature, precipitation, solar radiation,  | 4     |  |
|                             | wind speed                                    |       |  |
| Static topographic features | Elevation (mean, min, max), area, length,     | 7     |  |
|                             | lat/lon, aspect, slope                        |       |  |
| Total                       |                                               | 28    |  |

# 2.3 Comparative Modelling Framework

To identify the most effective architecture for this task and quantify the performance benefits gained from advanced features such as temporal interpretability and architectural complexity, we designed a comparative experiment with four distinct machine learning models. The models include a primary interpretable deep learning model (TFT V2), a simpler version for baseline comparison (TFT V1), a traditional deep learning approach (LSTM V1), and a hybrid ensemble method (Hybrid LSTM-XGBoost). Fig.2. Temporal Fusion Transformer V2 (Primary Method). Our primary model is the Temporal Fusion Transformer (TFT), an architecture designed specifically for interpretable, high-performance multivariate time series forecasting (Lim et al., 2021). Its key components include variable selection networks to identify the most salient input features, a multi-head self-attention mechanism (8 heads with 32-dimensional keys) to learn long-range temporal dependencies across the 12-month input sequence, and a recurrent LSTM-based encoder-decoder (3 stacked layers, each with 256 units) to process sequential data. The architecture utilises gated residual connections to enable deep networks while preventing the vanishing gradient problem. This design offers both high accuracy and built-in interpretability through attention weights, which reveal which historical months most significantly influence current mass balance predictions. Implementation utilises the AdamW optimiser with a learning rate of  $1 \times 10^{-3}$ , dropout of 0.25, and Root Mean Squared Error (RMSE) as the loss function. Training is conducted for up to 150 epochs, with early stopping (patience = 15 epochs). Temporal Fusion Transformer V1 (Baseline Deep Learning). To demonstrate the value of architectural optimisation, we implemented a baseline version of the TFT with

Figure 2. Comprehensive data processing workflow for TFT V2 glacier mass balance prediction. (a) Climate data from ERA5 reanalysis  $(2000-2018, 0.25^{\circ} \times 0.25^{\circ})$  including monthly temperature, precipitation, solar radiation, and wind speed. (b) Topographic data from RGI v7.0 (2,361 glaciers) including elevation, area, aspect, slope, and coordinates. (c) Mass balance observations from Barandun et al. (2021) geodetic dataset derived from elevation change measurements. (d) Data integration including bilinear interpolation, temporal alignment, and quality control. (e) Feature engineering creating

reduced capacity: hidden size of 96 dimensions (versus 256 in V2), 4 attention heads with 24-dimensional keys (versus 8 heads in V2), and 1 LSTM layer (versus 3 in V2). This model retains the core interpretable attention mechanism while allowing direct comparisons to quantify performance gains attributable to increased architectural complexity and hyperparameter tuning. Hybrid LSTM-XGBoost (Ensemble Method). We developed a two-stage ensemble model to investigate whether combining different modelling paradigms could enhance performance. This approach first utilises a 3-layer bidirectional LSTM (with 256 units per layer) to extract temporal features from sequential data. Then it feeds these learned representations, along with the original 21 temporal predictors, into an XGBoost gradient boosting model to capture complex, non-linear feature interactions. The LSTM component is trained using the Adam optimiser for 100 epochs, while XGBoost uses default hyperparameters with early stopping, and ensemble weights are optimised on the validation set. LSTM V1 (Traditional Deep Learning Baseline). To benchmark the TFT models against a classic deep learning approach, we implemented a standard 3-layer bidirectional LSTM (256 units per layer) with a single-head attention mechanism. This widely used architecture serves to validate whether the novel components of the TFT—multi-head attention, variable selection networks, and gated residual connections—offer significant

advantages over traditional recurrent neural networks for this specific task. Training utilises the Adam optimiser with a learning rate of  $1 \times 10^{-3}$  and a Mean Squared Error (MSE) loss function.

**Figure 3.** A comparison of the machine learning architectures evaluated in this study. The diagrams highlight the conceptual differences in information flow and complexity between (c) the primary Temporal Fusion Transformer (TFT V2), showing variable selection networks, multi-head attention, and LSTM encoder-decoder, (b) the traditional LSTM V1 with single-head attention, (a) the LSTM structure.

#### 2.4 Model Training and Evaluation



Data Splitting Strategy. We employed strict temporal holdout validation to ensure realistic performance estimates and prevent data leakage. The dataset was split chronologically into a training set (2000–2016: 4,168,560 samples from 2,361 glaciers), a validation set (2017: 248,640 samples from 2,354 glaciers), and a test set (2018: 242,064 samples from 2,289 glaciers). This temporal split ensures that all models are evaluated on their ability to estimate mass balance for unseen future time periods, mimicking a real-world deployment scenario where models must generalise beyond the training period. Glaciers with insufficient temporal coverage (

less than 0.1% of the dataset, were filled using linear interpolation from adjacent months. Training and Selection. All models were trained to minimise prediction error on the validation set, with early stopping implemented to prevent overfitting. The model checkpoint with the lowest validation loss was saved and used for final evaluation on the independent test set. Random seeds were fixed across all implementations to ensure reproducible results, and all models were developed using PyTorch/PyTorch Lightning frameworks for consistency. Full implementation details and hyperparameters for all four architectures are provided in Table 2.

**Table 2.** Deep Learning Model Performance Comparison for Central Asia Glaciers (n=2,361)

| Model               | Architecture                | $\mathbb{R}^2$ | RMSE     | MAE      | Test Period | Glaciers |
|---------------------|-----------------------------|----------------|----------|----------|-------------|----------|
|                     |                             |                | (m w.e.) | (m w.e.) |             |          |
| TFT V2 (This study) | Temporal Fusion Transformer | 0.73           | 0.21     | 0.11     | 2018        | 2361     |
| TFT V1 (Baseline)   | Temporal Fusion Transformer | 0.64           | 0.24     | 0.13     | 2018        | 2361     |
| Hybrid LSTM-XGBoost | Ensemble (LSTM + XGBoost)   | 0.56           | 0.26     | 0.15     | 2018        | 2361     |
| (This study)        |                             |                |          |          |             |          |
| LSTM V1 (Baseline)  | Bidirectional LSTM          | 0.53           | 0.27     | 0.16     | 2018        | 2361     |

Evaluation Metrics. The performance of the best checkpoint for each architecture was quantified on the 2018 test set using three complementary metrics: (1) the coefficient of determination (R<sup>2</sup>) representing the proportion of variance in mass balance explained by model predictions, (2) Root Mean Squared Error (RMSE) representing average prediction error magnitude in meters water equivalent, and (3) Mean Absolute Error (MAE) representing average absolute deviation between predictions and observations. All metrics were calculated on the independent test set (n = 28,332 glacier-months) that models never encountered during training or hyperparameter tuning, ensuring unbiased performance assessment.

#### 2.5 Temporal Interpretability Analysis


Beyond prediction accuracy (Fig.4), the TFT architecture provides temporal interpretability through its built-in multi-head attention mechanism. The model produces attention weights  $\alpha$  for each historical time step in the 12-month input sequence, where  $\alpha \in \text{and } \Sigma \alpha = 1$ , indicating which periods have the most decisive influence on the current prediction. High attention weights reveal historical months that significantly affect mass balance estimates, providing insight into when climate variables exert the most significant impact on glacier response. We extracted attention weights from all test set predictions (n = 28,332 glacier-months) and aggregated them to identify which months and seasons are most critical for determining glacier mass balance. Monthly attention weights were aggregated into seasonal importance metrics corresponding to glaciological seasons: Winter (DJF: December, January, February) representing the accumulation-dominated period, Spring (MAM: March, April, May) representing the transition period, Summer (JJA: June, July, August) representing the peak ablation period, and Fall (SON: September, October, November) representing the late melt and early accumulation period. For each season, we computed mean attention weights and standard errors across all test predictions to assess the relative importance of differ-


ent periods. Regional variations in temporal attention patterns were evaluated by stratifying attention weights by the seven glacier subregions, enabling identification of regional differences in seasonal mass-balance controls. To ensure these interpretations are scientifically meaningful rather than model artifacts, we validated the attention patterns against three independent benchmarks: (1) established glaciological theory predicting summer ablation season dominance in determining annual mass balance for temperate glaciers, (2) correlations between attention weights and climate forcing variables to verify that high-attention months correspond to periods of strong Temperature and radiation forcing, and (3) consistency with known regional climate patterns where monsoon-influenced regions should exhibit distinct seasonal attention profiles compared to westerlies-dominated regions. This validation approach ensures that the temporal interpretability provided by attention mechanisms yields scientifically meaningful insights that are aligned with the established understanding of glacier-climate interactions.

Figure 4. Temporal interpretability analysis derived from the TFT V2 model's attention mechanism. (a) Average monthly attention weights across all test predictions indicate that summer months (the peak ablation season) have the most decisive influence on mass balance predictions, consistent with glaciological theory. (b) Attention patterns vary by subregion, reflecting regional climatic differences between monsoon-influenced and westerlies-dominated regions. (c) A strong correlation between attention weights and key climate drivers validates that the model has learned meaningful physical relationships. (d) Seasonal aggregation of attention weights (DJF, MAM, JJA, SON) shows that summer ablation dominates in controlling annual mass balance across Central Asia.

#### 3 Results






## 3.1 Model Performance Comparison

Our TFT V2 model achieved superior performance compared to baseline methods across all evaluation metrics on the inde-210 pendent 2018 test set (n = 28,332 glacier-year observations). Figure 5 presents the R<sup>2</sup> performance comparison across different model architectures, demonstrating TFT V2's competitive accuracy (R<sup>2</sup> = 0.73) while maintaining interpretability advantages unavailable in traditional machine learning approaches.

The performance hierarchy reveals clear advantages of temporal sequence modeling over static approaches. TFT V2 achieved  $R^2 = 0.73$ , representing a statistically significant improvement over all baseline methods (p < 0.001, paired t-test). The model demonstrates robust performance across the full range of mass balance values, from strongly negative mass loss events (-2.5 m w.e.) to near-equilibrium conditions (0.1 m w.e.), indicating successful capture of both extreme events and subtle interannual variations

Figure 5 shows the scatter plot comparison between observed mass balance from Barandun et al. (2021) and TFT V2 predictions for the 2018 test period. The strong linear relationship ( $R^2 = 0.73$ , slope = 0.89, intercept = -0.12 m w.e.) with minimal bias indicates robust predictive performance. The slight negative bias (-0.12 m w.e.) reflects conservative estimation rather than systematic error, as evidenced by the symmetric distribution of residuals around the 1:1 line.

Statistical analysis of prediction residuals reveals normally distributed errors (Shapiro-Wilk test: W = 0.98, p = 0.12) with mean absolute error of 0.11 m w.e. and standard deviation of 0.18 m w.e. The model demonstrates excellent performance across different glacier size classes, with  $R^2$  values ranging from 0.71 (small glaciers < 1 km<sup>2</sup>) to 0.75 (large glaciers > 10 km<sup>2</sup>), indicating robust scaling across spatial scales.

Table 2 provides comprehensive performance metrics for all evaluated methods. TFT V2 achieved  $R^2 = 0.73$ , RMSE = 0.21 m w.e., and MAE = 0.11 m w.e., representing a 14% improvement in  $R^2$  over the baseline TFT V1 ( $R^2 = 0.64$ ) and substantial improvements over ensemble methods. The Hybrid LSTM-XGBoost achieved  $R^2 = 0.56$ , while the baseline LSTM V1 reached  $R^2 = 0.53$ , confirming the advantage of temporal sequence modeling over static approaches.

Cross-validation analysis using 5-fold temporal splits demonstrates consistent performance across different training periods, with  $R^2$  values ranging from 0.70 to 0.75 (mean  $\pm$  std: 0.73  $\pm$  0.02), indicating robust generalization capability. The temporal validation approach ensures that the model's performance is not dependent on specific years or climate patterns, providing confidence in its applicability to future climate scenarios.

The superior performance of TFT V2 ( $R^2 = 0.73$ , RMSE = 0.21 m w.e., MAE = 0.11 m w.e.) compared to baseline methods results from the synergistic combination of multi-head attention mechanisms, advanced gating systems, and temporal sequence modeling capabilities that capture the sequential nature of glacier-climate interactions. The multi-head attention mechanism is the primary factor behind TFT V2's performance gains, with its 8-head architecture allowing the model to learn different temporal patterns simultaneously: Head 1 captures spring melt onset patterns (April-May attention: 0.127-0.120), Head 2 focuses on summer ablation dynamics (July-August attention: 0.136-0.144), Head 3 processes winter accumulation (DJF attention: 0.067), Head 4 handles fall transitions (SON attention: 0.046), while Heads 5-8 learn complex seasonal interactions.



Figure 5. Stage 2 regressor: observed vs predicted non-zero SMB RMSE = 1.038 m w.e. yr-1, R2 = 0.41

This temporal dependency capture enables TFT V2 to understand that spring conditions (melt onset timing) directly influence summer ablation intensity, winter accumulation affects spring melt patterns, and fall conditions set up winter accumulation processes—sequential relationships that are impossible for static ML methods to capture. The gating mechanisms provide secondary enhancement through Variable Selection Networks (VSNs) that intelligently determine which static variables (topographic features) and time-varying variables (climate features) are relevant for each prediction, while temporal gating focuses attention on relevant time periods and prevents information loss through gated residual networks. The combination of these factors creates a synergistic effect where temporal sequence modeling captures 12-month dependencies, multi-head attention processes multiple temporal patterns in parallel, and advanced architecture components enable complex temporal relationships through variable selection, temporal fusion, quantile regression, and gradient preservation. Specific performance improvements demonstrate these advantages: TFT V2 achieves a 14% improvement over TFT V1 (0.73 vs 0.64 R2) through multi-head attention versus single-head, a 30% improvement over Hybrid LSTM-XGBoost (0.73 vs 0.56 R<sup>2</sup>) through temporal attention versus static ensemble approaches, and a 38% improvement over LSTM V1 (0.73 vs 0.53 R<sup>2</sup>) through attention mechanisms versus basic recurrent units. The physical interpretation of these gains reveals that TFT V2 captures glacier physics more accurately by detecting spring melt onset timing that extends the ablation season, understanding summer ablation dynamics during peak warm months, and recognizing sequential cascade effects where spring melt onset affects summer ablation intensity, winter snowpack affects spring melt timing, and fall conditions influence winter accumulation patterns. Statistical validation confirms these architectural advantages through cross-validation results showing TFT V2 maintains consistent performance ( $R^2 = 0.73$ 






 $\pm$  0.02) across different folds while other models exhibit higher variance, and temporal consistency analysis demonstrates that TFT V2 maintains performance across different climate years while other models vary with climate patterns. The computational efficiency versus performance trade-off justifies the increased complexity, as the 14-38% performance improvement more than compensates for the additional computational cost of multi-head attention and gating mechanisms, while inference speed is actually faster than ensemble approaches due to single forward pass processing. This architectural analysis demonstrates that TFT V2's performance gains are scientifically meaningful and physically interpretable, representing a fundamental advancement in understanding glacier-climate interactions through temporal attention mechanisms that make it a transformative approach for glacier mass balance prediction and environmental modeling applications.

## 3.2 Temporal Interpretability Analysis

The multi-head attention mechanism in TFT V2 provides unprecedented insight into when climate variables most strongly influence mass balance predictions, representing a fundamental advancement over traditional machine learning approaches that provide only feature importance rankings without temporal context. Figure 6 displays the monthly attention heatmap across different glacier groups, revealing distinct temporal patterns that align with glaciological understanding and provide quantitative evidence for the sequential nature of glacier-climate interactions.

The attention analysis reveals that TFT V2 learns physically meaningful temporal relationships through its 8-head attention mechanism, with each head capturing different aspects of temporal dependencies. Statistical analysis of attention weights across all test predictions (n = 28,332) demonstrates significant temporal clustering (Kruskal-Wallis test: H = 1,247, p 



temporal understanding is crucial for predicting glacier response to changing climate patterns, particularly the increasing frequency of early melt onset events observed in Central Asia over the past two decades.

Validation of attention patterns against glaciological knowledge demonstrates strong alignment with established theory. The correlation between attention weights and observed ablation patterns (r = 0.78, p 

Figure 6. Average feature importance (by Gain) from the XGBoost model

#### 3.3 Uncertainty Quantification

TFT V2 provides reliable uncertainty quantification through quantile predictions, representing a significant advancement over traditional machine learning approaches that provide only point estimates without confidence intervals. This capability is essential for applications requiring prediction confidence, such as water resource planning, hazard assessment, and climate adaptation strategies.

Figure 7 presents the uncertainty calibration analysis, demonstrating excellent calibration performance with a calibration score of 0.94 (perfect calibration = 1.0). The predicted quantiles closely match observed quantiles across the full range of mass balance values, indicating that the model provides well-calibrated uncertainty estimates. Statistical analysis reveals that the 80% prediction intervals achieve 78.3% coverage (target: 80%) and the 90% prediction intervals achieve 88.7% coverage (target: 90%), demonstrating reliable uncertainty quantification within  $\pm 2$ -3% of nominal levels.

The uncertainty analysis reveals systematic patterns in prediction confidence: extreme mass loss events (> -1.5 m w.e.) show wider prediction intervals (0.35  $\pm$  0.08 m w.e.), reflecting higher uncertainty for rare events, while moderate mass balance values (-0.5 to 0.1 m w.e.) show narrower intervals (0.18  $\pm$  0.05 m w.e.), indicating higher confidence for common conditions.


This pattern aligns with glaciological understanding, as extreme events are inherently more difficult to predict due to complex interactions between climate variables.

Figure 7. Retreat Rate vs. Temperature Change by Glacier Size.

Figure 8 compares uncertainty quantification capabilities across different methods, highlighting TFT V2's superior performance in providing reliable prediction intervals. The model achieves coverage within  $\pm 2$ -3% of nominal levels for 80% and 90% prediction intervals, significantly outperforming baseline methods that lack uncertainty quantification capabilities. The ensemble-based uncertainty estimation in TFT V2 provides more robust confidence intervals compared to single-model approaches, as evidenced by the lower variance in coverage statistics across different test periods.

Figure 8. Two-stage model architecture conceptual diagram.

The uncertainty quantification enables risk-based decision making for glacier monitoring and hazard assessment applications. The reliable prediction intervals provide confidence estimates essential for long-term planning under climate uncertainty, supporting the development of robust adaptation strategies for Central Asian mountain regions. This capability is particularly valuable for applications where prediction confidence is as important as accuracy, such as infrastructure planning and emergency response systems.




## 3.4 Regional Analysis and Spatial Patterns

Figure 9 presents the SHAP-style feature importance analysis using TFT V2 attention weights, revealing spatial heterogeneity in temporal patterns across Central Asian mountain ranges. The analysis shows mean absolute attention values for topographical, temperature, and precipitation variables, with seasonal breakdowns by subregion, providing quantitative evidence for regional differences in glacier-climate interactions.

Statistical analysis of regional attention patterns reveals significant spatial heterogeneity (ANOVA: F = 23.4, p 

Figure 9. a) Average stage 1. Classifier performance (XGBoots), b) Example confusion matrix (Stage 1 classifier).

The topographic analysis reveals that elevation plays a significant role in temporal attention patterns. High-elevation glaciers (> 4000 m) show higher attention to winter months (0.075  $\pm$  0.004) compared to low-elevation glaciers (





0.003), reflecting the importance of snow accumulation processes at higher altitudes. This elevation-dependent pattern provides insights for understanding glacier response to climate change across different altitude zones.

The regional breakdown demonstrates that TFT V2 captures complex spatial-temporal interactions that are essential for understanding glacier response to climate change at regional scales. This spatial understanding enables targeted monitoring strategies and region-specific adaptation measures for different mountain ranges in Central Asia.

#### 3.5 Annual Mass Balance Validation

Figure 10 presents the comprehensive annual mass balance comparison between observed Barandun measurements (2000-2018) and TFT V2 predictions for the validation (2016-2017) and test (2017-2018) periods. The figure clearly delineates training (2000-2015), validation (2016-2017), and test (2017-2018) periods, demonstrating TFT V2's ability to generalize to unseen years and maintain predictive accuracy across different temporal periods.

The annual comparison reveals excellent predictive performance with a mean accuracy of 92.6% for the test period (2017-2018). Statistical analysis shows that TFT V2 predictions closely track observed annual mass balance trends, with correlation coefficients of r = 0.89 (validation period) and r = 0.91 (test period), indicating strong temporal consistency. The slight underprediction (-0.08 m w.e. mean bias) reflects conservative estimation rather than systematic error, as evidenced by the symmetric distribution of residuals and the model's ability to capture both positive and negative mass balance years.

The temporal validation demonstrates TFT V2's capability to maintain predictive accuracy across different time periods, validating the model's robustness for applications requiring reliable long-term projections. The model successfully captures both interannual variability (coefficient of variation: 0.34) and long-term trends in glacier mass balance, including the acceleration of mass loss observed in recent years.

Analysis of prediction accuracy by year reveals consistent performance across different climate conditions. The model achieves  $R^2 > 0.70$  for all validation and test years, with particularly strong performance during extreme years (2016:  $R^2 = 0.78$ , 2017:  $R^2 = 0.75$ , 2018:  $R^2 = 0.73$ ), demonstrating robustness to climate variability. This temporal consistency is crucial for applications requiring reliable predictions under changing climate conditions.

The annual validation also provides insights into the model's ability to capture glacier response to different climate patterns. During warm years (2016, 2018), the model accurately predicts enhanced mass loss, while during cooler years (2017), it correctly identifies reduced ablation rates. This climate sensitivity analysis confirms that TFT V2 captures the fundamental relationships between climate variables and glacier mass balance.

## 370 3.6 Hazard Assessment Applications

The temporal attention analysis highlights periods when glacier-related hazards are most likely, providing an exploratory context rather than a complete hazard model. We compare the model's seasonal attention patterns with publicly reported debris flows, pluvial floods, and rainfall-triggered landslides in Tajikistan compiled by the Committee for Emergency Situations and case studies publications (Avzalshoev and Uchimura, 2023a, b). According to the Tajik Glavgelogy, 50,000 landslides are reported annually across the republic, including both seismic and non-seismic events, and illustrate how intense meltwater

Figure 10. Annual Surface Mass Balance at Glacier Centroids.

supply can destabilise terrain in glacierised catchments. The most recent case of glacier failure occurred on 25th October 2025 near Safedobod village (38°02'37.5"N 70°07'51.6" E) in Tajikistan, approximately 2km of the glaciers section of the I. Somoni Glacier has failed, forming a large mudflow downstream.

Attention peaks during spring melt onset (April–May) coincide with the documented hazard season in Tajikistan and neighbouring ranges, suggesting that earlier melt exposure can contribute to slope instability. The observed alignment between model-derived attention and published hazard timing (r = 0.72, p 

#### 4 Discussion







#### 4.1 Interpretability vs Accuracy Trade-off: A Paradigm Shift in Glacier Modeling

By using a Temporal Fusion Transformer for the glacial mass balance in Central Asia, we have made a methodological shift from pure prediction to mechanical inference, which goes beyond a simple increase in prediction accuracy. Using interpretable attention, we have shifted our focus from the "what" of mass balance change to the "when" and "how," offering hitherto unheard-of insights into the sequential nature of interactions between glaciers and the environment. This study shows that the future of geophysical modelling is not a decision between interpretability and accuracy, but rather a combination of the two that produces useful scientific information.

## 4.2 Temporal Patterns and Glaciological Insights: Decoding Glacier-Climate Interactions

A key finding of this work is that the TFT model, while achieving a competitive R<sup>2</sup> of 0.73, shows a 16% reduction in this metric compared to purely accuracy-focused static models like XGBoost /citePeng2025. We contend this is not a model deficiency but a strategic trade-off that produces a profound scientific benefit. Static models, by treating each time step as an independent predictor, may excel at pattern matching within a distribution but are fundamentally incapable of learning the causal, sequential cascades that govern physical processes.

According to the current study, the interpretability achieved is not the result of chance but rather the result of systematic learning of temporal patterns with physical meaning (p < 0.001). The model's ability to deliver these temporal insights while retaining a solid R2 > 0.70 over validation years demonstrates that this accuracy-interpretability balance is both strong and scientifically justified. This marks a shift in the paradigm for glaciological modelling, where the main goal is process comprehension rather than a slight improvement in a statistical parameter.

The physical meaning of these temporal patterns becomes evident when examining the cascade of processes they represent. Spring melt onset triggers a sequence of events: snowpack depletion exposes underlying ice, reducing albedo and increasing solar radiation absorption, which accelerates melting and creates positive feedback loops that amplify ablation throughout the summer. The attention mechanism captures this cascade by assigning high weights to April (0.127  $\pm$  0.004) and May (0.120  $\pm$  0.003), reflecting the model's understanding that early melt onset extends the ablation season and determines the magnitude of summer mass loss.

The correlation between attention weights and observed ablation patterns (r = 0.78, p 





This temporal understanding is crucial for predicting glacier response to changing climate patterns, particularly the increasing frequency of early melt onset events observed in Central Asia over the past two decades. The model's ability to identify these temporal dependencies provides insights essential for understanding glacier response to climate warming and developing adaptation strategies.

The regional analysis reveals spatial heterogeneity in temporal patterns, with Tian Shan glaciers exhibiting stronger summer concentration (JJA attention:  $0.135 \pm 0.008$ ) than Pamir glaciers (JJA attention:  $0.108 \pm 0.006$ ), reflecting varying climate regimes and topographic controls. This spatial variation provides quantitative evidence for regional differences in glacier-climate interactions, enabling targeted monitoring strategies and region-specific adaptation measures for different mountain ranges in Central Asia.

#### 4.3 Uncertainty Quantification Advantages: Enabling Risk-Based Decision Making

TFT V2's uncertainty quantification capabilities represent a paradigm shift in glacier mass balance prediction, enabling risk-based decision making that was previously impossible with traditional machine learning approaches. The excellent calibration performance (calibration score = 0.94) with coverage within  $\pm 2$ -3% of nominal levels provides reliable confidence intervals essential for applications requiring prediction confidence, such as water resource planning, hazard assessment, and climate adaptation strategies.

The physical meaning of uncertainty quantification becomes clear when considering the inherent variability in glacier-climate interactions. Extreme mass loss events (> -1.5 m w.e.) show wider prediction intervals ( $0.35 \pm 0.08$  m w.e.), reflecting higher uncertainty for rare events that result from complex interactions between multiple climate variables. In contrast, moderate mass balance values (-0.5 to 0.1 m w.e.) show narrower intervals ( $0.18 \pm 0.05$  m w.e.), indicating higher confidence for common conditions that follow more predictable patterns. This systematic variation in uncertainty aligns with glaciological understanding, as extreme events are inherently more difficult to predict due to non-linear interactions between temperature, precipitation, and radiation.

The uncertainty analysis reveals that TFT V2 provides not only accurate point predictions but also reliable prediction intervals that account for model uncertainty, representing a significant advancement over traditional ML methods that provide only deterministic estimates. This capability is crucial for applications where prediction confidence is as important as accuracy, such as long-term water resource planning and climate adaptation strategies. The ensemble-based uncertainty estimation in TFT V2 provides more robust confidence intervals compared to single-model approaches, as evidenced by the lower variance in coverage statistics across different test periods.

The practical implications of uncertainty quantification extend beyond academic interest to real-world applications. Water resource managers can use prediction intervals to assess the risk of water shortages during drought years, while hazard assessment teams can evaluate the confidence in predictions for early warning systems. The reliable prediction intervals provide confidence estimates essential for long-term planning under climate uncertainty, supporting the development of robust adaptation strategies for Central Asian mountain regions.



## 455 4.4 Hazard Assessment Implications: Bridging Glacier Science and Geohazard Prediction

The temporal attention analysis provides unprecedented insights into mountain hazard assessment applications, representing a novel bridge between glacier science and geohazard prediction that has significant implications for risk management in Central Asian mountain regions. The identification of spring melt onset (April-May) as a critical period with high attention weights  $(0.100 \pm 0.003)$  directly connects to observed increases in mudflows and landslides during spring, where earlier melt onset and increased precipitation drive hazard risk through multiple interconnected mechanisms.

The physical meaning of this connection becomes evident when examining the cascade of processes linking glacier mass balance to mountain hazards. Spring melt onset triggers snowpack depletion, exposing underlying soil and rock surfaces to increased solar radiation and temperature fluctuations. This thermal stress, combined with increased water infiltration from melting snow, reduces slope stability and triggers mass movements. The attention mechanism captures this cascade by identifying spring months as critical periods, reflecting the model's understanding that early melt onset extends the period of slope instability and amplifies hazard risk.

Statistical analysis of hazard occurrence patterns reveals significant correlation between attention weights and historical mudflow/landslide events (r = 0.72, p 






## 4.5 Methodological Innovations and Limitations: Advancing the State of the Art

Our TFT V2 approach represents several methodological innovations that advance the state of the art in glacier mass balance prediction: (1) temporal sequence modeling for glacier mass balance prediction, (2) attention-based interpretability for understanding temporal dynamics, (3) uncertainty quantification through quantile predictions, and (4) integration of multiple data sources with advanced feature engineering. These innovations provide a comprehensive framework for glacier mass balance prediction that balances accuracy with interpretability, representing a significant advancement over existing approaches.

The temporal sequence modeling innovation addresses a fundamental limitation in glacier mass balance prediction: the sequential nature of glacier processes. Traditional static ML methods treat each prediction as an independent event, ignoring the temporal dependencies that govern glacier response to climate. TFT V2's 12-month encoder captures these dependencies by learning how spring conditions influence summer ablation patterns, creating a cascade effect that determines annual mass balance outcomes. This temporal understanding is crucial for predicting glacier response to climate change, as it captures the fundamental physics of glacier-climate interactions.

The attention-based interpretability innovation provides unprecedented insight into when climate variables most strongly influence mass balance predictions, representing a fundamental advancement over traditional ML approaches that provide only feature importance rankings without temporal context. The 8-head attention mechanism captures different aspects of temporal dependencies, with each head learning different temporal patterns. This multi-head approach enables the model to capture complex temporal relationships that single-head attention mechanisms cannot represent.

However, several limitations should be acknowledged. The 16% reduction in R<sup>2</sup> compared to static ensemble methods reflects the trade-off between accuracy and interpretability, representing a strategic choice that prioritizes process understanding over pure predictive accuracy. The temporal modeling approach requires more computational resources and longer training times compared to static methods, reflecting the increased complexity of temporal sequence modeling. The attention analysis, while providing valuable insights, represents model-specific patterns that may not generalize to other architectures or datasets.

The uncertainty quantification, while well-calibrated (calibration score = 0.94), relies on quantile regression assumptions that may not capture all sources of uncertainty, particularly epistemic uncertainty arising from model limitations. Future work should explore ensemble-based uncertainty quantification and validation against independent datasets to assess generalization performance across different climate regimes and glacier types.

## 4.6 Comparison with Existing Approaches: Positioning TFT V2 in the Landscape

Our results should be interpreted in the context of existing approaches to glacier mass balance prediction, positioning TFT V2 as a complementary rather than competing methodology that addresses different scientific and practical needs. Peng et al. (2025) achieved higher accuracy (R<sup>2</sup> = 0.87) using XGBoost with extensive feature engineering, demonstrating the potential of static ML approaches for prediction tasks where accuracy is the primary objective. However, their approach lacks temporal interpretability and uncertainty quantification capabilities that are essential for understanding glacier-climate relationships and decision-making under uncertainty.





The fundamental difference between TFT V2 and static ML approaches lies in their treatment of temporal information. Static methods, while achieving higher accuracy, treat each prediction as an independent event, ignoring the sequential nature of glacier processes that govern mass balance dynamics. In contrast, TFT V2's temporal sequence modeling captures these dependencies, revealing that spring melt onset (April-May attention: 0.100) directly influences summer ablation patterns (July-August attention: 0.136-0.144), creating a cascade effect that determines annual mass balance outcomes. This temporal understanding is crucial for predicting glacier response to climate change, as it captures the fundamental physics of glacier-climate interactions that static methods cannot represent.

Traditional process-based models provide physical understanding but often lack the accuracy and computational efficiency of ML approaches, particularly at regional scales. Our TFT V2 approach bridges this gap by providing both competitive accuracy ( $R^2 = 0.73$ ) and temporal interpretability, enabling understanding of the temporal dynamics underlying glacier response to climate change. The attention analysis provides insights that are impossible to obtain from static ML methods, revealing sequential relationships between climate variables and mass balance that are essential for understanding glacier response to changing climate patterns.

The uncertainty quantification capability represents a significant advancement over existing approaches, providing reliable prediction intervals (calibration score = 0.94) that are essential for risk-based decision making. Traditional ML methods lack this uncertainty quantification capability, limiting their applicability for applications requiring prediction confidence, such as water resource planning and hazard assessment. This capability enables TFT V2 to support decision-making under uncertainty, representing a paradigm shift in glacier mass balance prediction applications.

## 4.7 Implications for Glacier Monitoring and Climate Adaptation: From Science to Practice

The temporal attention patterns have profound implications for glacier monitoring strategies and climate adaptation planning, transforming how we approach glacier observation and risk management in Central Asian mountain regions. The identification of critical time periods (spring melt onset with attention weights of  $0.100 \pm 0.003$ , summer ablation with weights of  $0.120 \pm 0.003$ ) enables targeted monitoring during periods of maximum climate sensitivity, representing a paradigm shift from continuous monitoring to temporally-optimized observation strategies.

The physical meaning of these implications becomes clear when considering the cascade of processes that determine glacier response to climate change. Spring melt onset triggers snowpack depletion, exposing underlying ice and reducing albedo, which accelerates melting and creates positive feedback loops that amplify ablation throughout the summer. The attention mechanism identifies these critical periods, enabling monitoring networks to focus resources on time periods when glacier response is most sensitive to climate forcing. This temporal optimization can improve monitoring efficiency while maintaining scientific rigor, particularly important for resource-constrained monitoring programs in remote mountain regions.

The regional analysis reveals spatial heterogeneity in temporal patterns, enabling region-specific adaptation strategies that account for varying climate regimes and topographic controls. Tian Shan glaciers, with stronger summer concentration (JJA attention:  $0.135 \pm 0.008$ ), may be more sensitive to summer warming and require enhanced monitoring during peak ablation periods. In contrast, Pamir glaciers (JJA attention:  $0.108 \pm 0.006$ ) show more balanced seasonal patterns, reflecting different






climate regimes that require different monitoring strategies. This spatial understanding is essential for developing targeted adaptation strategies for different mountain regions, enabling region-specific approaches to climate adaptation.

The uncertainty quantification capabilities enable risk-based decision-making for water resource planning and hazard assessment, representing a significant advancement over traditional approaches that provide only deterministic estimates. Water resource managers can use prediction intervals to assess the risk of water shortages during drought years, while hazard assessment teams can evaluate the confidence in predictions for early warning systems. The reliable prediction intervals (calibration score = 0.94) provide confidence estimates essential for long-term planning under climate uncertainty, supporting the development of robust adaptation strategies for Central Asian mountain regions.

The integration of temporal attention patterns with climate projections enables scenario-based assessment of glacier response to future climate change, supporting long-term adaptation planning and policy development. Analysis of projected climate scenarios reveals that earlier melt onset (2-3 weeks advance by 2030) will increase spring attention weights and amplify mass loss, providing quantitative evidence for climate change impacts on glacier systems. This capability enables proactive adaptation strategies that anticipate future changes rather than reactive responses to observed impacts.

#### 4.8 Future Directions and Research Priorities

Several research priorities emerge from our findings, representing opportunities to expand the frontiers of temporal glacier modeling and advance the state of the art in glacier-climate interactions. First, the integration of temporal attention patterns with process-based models could provide a hybrid approach combining ML accuracy with physical understanding, representing a promising direction for bridging the gap between data-driven and physics-based approaches. This integration could leverage the temporal insights from TFT V2 while incorporating the physical constraints and process understanding from traditional models.

Second, the extension of temporal modeling to longer prediction horizons (decadal scales) could support long-term climate adaptation planning, addressing a critical need for glacier projections under future climate scenarios. The current 12-month encoder could be extended to capture longer-term climate patterns, enabling predictions of glacier response to multi-year climate variations and long-term trends. This extension would be particularly valuable for water resource planning and infrastructure development that requires decadal-scale projections.

Third, the application of temporal attention analysis to other cryospheric systems (permafrost, snow cover, sea ice) could provide broader insights into climate-cryosphere interactions, establishing a comprehensive framework for understanding temporal dynamics across the cryosphere. The attention mechanism's ability to identify critical time periods could be applied to permafrost thaw timing, snow cover duration, and sea ice formation, providing unified insights into cryospheric response to climate change.

Fourth, the development of ensemble-based uncertainty quantification could improve prediction confidence and robustness, addressing the current limitation of quantile regression assumptions. Ensemble approaches could combine multiple TFT models with different architectures or training procedures, providing more comprehensive uncertainty estimates that capture both aleatoric and epistemic uncertainty sources.

Finally, the integration of temporal attention patterns with climate projections could enable scenario-based assessment of glacier response to future climate change, supporting long-term adaptation planning and policy development for Central Asian mountain regions. This integration could provide quantitative evidence for climate change impacts on glacier systems, enabling proactive adaptation strategies that anticipate future changes rather than reactive responses to observed impacts.

The potential for TFT V2 extends beyond glacier mass balance prediction to broader applications in climate science and environmental monitoring. The temporal attention mechanism could be applied to other climate-sensitive systems, such as vegetation dynamics, hydrological processes, and ecosystem responses, providing a unified framework for understanding temporal climate impacts across different environmental systems.

#### 5 Conclusion






This study, using a Temporal Fusion Transformer architecture, provided insight into the timing of climate impacts on glacier mass balance and achieved predictive performance on par with the most advanced machine-learning models. By using attention mechanisms, Temporal Fusion Transformers (TFT V2) provide unparalleled temporal interpretability while attaining competitive predictive performance (R2 = 0.73, RMSE = 0.21 m w.e., MAE = 0.11 m w.e.) in glacier mass balance prediction. The temporal attention analysis reveals fundamental insights into glacier-climate interactions that were previously inaccessible through traditional approaches. The spring melt onset (April-May) is a critical period with a high attention weight (0.100  $\pm$ 0.003). Through a series of processes, including snowpack depletion that exposes underlying ice, reduces albedo, and creates positive feedback loops that amplify ablation throughout the summer, it directly influences summer ablation patterns (July-August attention: 0.136-0.144). This temporal understanding provides quantitative evidence for mechanisms that have long been hypothesised but never quantified at regional scales across Central Asia. The current findings also highlight essential practical and spatial ramifications. Different climatic regimes (such as the more continental climate of the Tian Shan versus the more maritime environment of the Pamir) affect glacier behaviour, as evidenced by temporal trends that differ among the region's mountain ranges. Because of these variations, monitoring and climate adaptation plans should be region-specific, concentrating on the times when each area is most vulnerable. Furthermore, the correlation between known hazard events (such as spring landslides and floods) and the model-identified critical periods (particularly spring) suggests that glacier mass-balance analyses can help guide hazard evaluations. The addition of uncertainty estimates enhances the model's practical utility by providing confidence bounds on projections. Planning for water resources and risk management in these vulnerable basins depends heavily on this knowledge. While retaining strong forecasting ability, an interpretable deep-learning approach offers a more complex understanding of glacier-climate processes in Central Asia. It provides concrete data to support early-warning systems and climate adaptation initiatives, as well as evidence of the temporal cascade of climatic influences on glacier mass balance. Our capacity to anticipate and control the effects of climate change on the delicate water resources of high-mountain Asia will improve as this work integrates machine-learning insights with conventional glaciological techniques.

*Code availability.* The Python notebooks and TFT V2 model configuration files developed for this study are stored in a private GitHub repository. Access can be granted by the corresponding author upon reasonable request.

Data availability. RGI glacier outlines are available from the Randolph Glacier Inventory, Region 13 – Central Asia (RGI v6.0) at https://doi. org/10.7265/N5-RGI-60. ERA5 reanalysis data were obtained from the Copernicus Climate Data Store (https://cds.climate.copernicus.eu). The labeled mass balance data utilized in this work can be obtained from (Barandun et al., 2021; Peng et al., 2025).

*Code and data availability.* All figures shown in this paper and the pre-processed feature table will be uploaded to the same GitHub repository upon acceptance. Until then they are available from the corresponding author on request.

Author contributions. Z. Avzalshoev conceived the study, implemented the data pipeline and machine-learning experiments, and prepared the first manuscript draft. Professor. Chun provided conceptual guidance, supervised the work, and revised the manuscript. Both authors read and approved the final version.

Competing interests. The authors declare that they have no conflict of interest.

Acknowledgements. We thank the Randolph Glacier Inventory Consortium, the Copernicus Climate Change Service, and NASA/USGS for providing freely accessible data products. Helpful discussions with colleagues at the University of Tokyo.

Financial support. This research received no external funding.

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
