# Peer review of "Interpretable Deep Learning for Glacier Mass Balance: Temporal Attention Patterns in Central Asia"

_EGUsphere, 2025_

## Referee Comment (RC1)

**Review of "Interpretable Deep Learning for Glacier Mass Balance: Temporal Attention Patterns in Central Asia", by Zafar Avzalshoev and Pang-jo Chun**

EGUsphere

Jordi Bolibar and Kamilla H. Sjursen

**1 General comments**

Avzalshoev and Chun present a study where they apply different machine learning (ML) methods, including Temporal Fusion Transformers (TFT V2), to simulate surface mass balance of glaciers in Central Asia. While the paper introduces some interesting concepts, like new methods for modelling surface mass balance or attempting to relate it to glacier-related hazards, the methods have fundamental flaws that discard the validity of the results. Importantly, the methodology and results do not support the objectives and conclusions stated in the paper. In our opinion, the problems are so fundamental that a complete revision of the study is necessary for the results and arguments to be valid. Therefore, we recommend rejecting the paper and encourage the authors to re-submit it when the fundamental problems in the methods have been addressed.

Here we will go over the main reasons why we recommend a rejection and what should be improved in the methods in order to draw scientific conclusions from them. We will address these points in different general comments (GCs).

**1.1 GC1: Training a machine learning model on the outputs of another model**

The first fundamental flaw of this paper is the fact that the authors claim that they train their ML models on "geodetic mass balance data" (e.g. L7-8, L74, L94-96, or "measurements" on L349). However, these training data are in fact the output of a temperature-index model calibrated on geodetic mass balance and snowlines (Barandun et al., 2021). This strongly influences the conclusion that can be drawn from their work and creates a misalignment between the research objectives, methodology and analysis. Training an ML model on the outputs of another model can indeed make sense in some cases, such as when one intends to emulate an expensive physical model at a fraction of the cost by "compressing" it using statistical learning. That is not the case here. Barandun et al. (2021) already performed the task that the authors intend to do in this study, that is, calibrating a surface mass balance model based on different

types of observations to reconstruct the full surface mass balance series of the whole region for a period of time. The authors here are emulating a temperature-index model, which is a very simple model with only two parameters, by using a rather complex deep learning model with thousands of parameters. This doesn't make sense from a scientific point of view, because the primary reason one would emulate a physical model is to make it faster, which is not the case here. For the study to make scientific sense and to provide added value, the authors could instead try to replicate what Barandun et al. (2021) did in their study, and try to improve the accuracy of a model by training on the same observations (i.e. the snowlines and the geodetic mass balance) as target data. That would provide an added value, and results could be compared to the ones of a temperature-index model to see if any extra physical processes or nonlinearities are being captured by the more complex deep learning model. With the use of modelled data it is also unclear if this approach is feasible for real-world observational data, which is noisy and sparse. The reported performance is not representative for such data and it is unclear whether the insights gained from the TFT actually provide new insights or just an interpretation of the "inner workings" of the temperature-index model.

In light of the current goals of the paper, we recommend the authors to rethink their study and to train the model **only** based on observations, and not based on the outputs of other models.

**1.2  GC2: Fundamental statistical issues in the training**

The second fundamental flaw we see in this paper concerns several statistical problems related to the training of the model, which strongly damage its validity and performance. The main problems are related with how the dataset is divided into train, validation and test, which crucially determines what the model will learn. The authors mention that their intention is to make the model capable of making future predictions in time, so they perform a temporal split. The fact that they chose to use just a single year for validation and a single year for test is clearly not enough. Glacier surface mass balance is characterized by interannual variability linked to the forcing climate. If the performance of the model is assessed on a single year (both for validation for training and test for independent performance assessment) it is impossible to evaluate if the model has learnt to capture that interannual variability. The claims that the model demonstrates accuracy over different time periods and climatic conditions and is robust for long-term predictions ( Section 3.5 and Fig. 10) are therefore not supported by the test performance. Moreover, if the chosen hydrological year happens to have a climate very close to the median, the model model performance will come across as much better than if that year represents an outlier with extreme conditions.

This has several implications for the study. First, Figure 10 clearly shows that the model is not working, with poor performance for the two years that are evaluated (validation and test). This shows that metrics alone do not provide a complete picture, and the model likely does not capture the right physics. Additionally, the benefits of moving to larger and more complex models such as TFTs are reported only in terms of $R^2$, which is generally not a good metric for these cases. This is even more problematic taking into account that the metrics are computed based on a single year. This means that $R^2$ is just capturing the spatial variability between the surface mass balance of glaciers across regions, rather than the interannual variability, which

was reported to be the main goal of the modelling. Considering the other reported metrics (i.e. RMSE and MAE), we can see almost no added value in using these more complex models. This indicates that there is limited learning, perhaps because the training procedure is flawed. It would be interesting to see the training vs validation errors to assess if the model(s) are overfitting or underfitting.

Finally, the claims about model interpretability linked to temporal attention are not true. The methods used to assess the importance or contribution of features to given years or the global performance of the model can be assessed for any of the other models, using tools such as SHAP values. Moreover, tree-based methods are generally known to be much more interpretable than neural networks, and the same analysis of understanding the importance of individual months or features could also be obtained with the right design of the model and the training process.

**1.3 GC3: Link to glacier hazards**

The connection between the results of the ML models and glacier hazards has limited scientific explanation or statistical grounds, such that the strong statements regarding this link (e.g. Section 4, L12-14) are not sufficiently supported. Importantly, the statistical results presented are not sufficiently explained and based on unpublished data without reference (Section 3.6). The ML model is meant to predict glacier surface mass balance, which per se is not necessarily directly linked to glacier-related hazards. The authors use the attention weights from the model predicting surface mass balance to argue that they can also indicate glacier-related hazards. This connection is weak and not sufficiently supported by evidence. The attention weights are only indications of the contribution or importance of those months to explain/predict surface mass balance, not hazards.

**1.4 GC4: Overall quality of the figures and references**

A bit less relevant, but also important, is the overall quality of the figures. Many of them are difficult to read, with panels overlapping text and other basic issues which hamper their readability and the communication of the results. Moreover, the authors mention that there is supposed to be a scatter plot in Figure 5, which is not the case. A direct comparison with the ground truth data is never shown.

Several statements lack references (e.g. L278-279, L333-335 and L338-340) and the discussion section fails to relate the findings of the study to the current state of research (a single reference is mentioned in the six page discussion) and lacks a discussion of the limitations of the approach. In some places references do not support the statements (e.g. it is not clear why Huss et al. (2008) and Rastner et al. (2019) are referenced on L42 for geodetic mass balance, and why Farinotti et al. (2019) is referenced as an example of machine learning to estimate ice thickness on L48?).

**1.5 GC5: Closed code**

Finally, the authors do not seem to be willing to open-source their code. We understand that they might not want to share it until the paper is published, but open-sourcing scientific work,

including code, should be a must.

---

## Referee Comment (RC2)

Review of

„Interpretable Deep Learning for Glacier Mass Balance: Temporal Attention Patterns in Central Asia"

by Zafar Avzalshoev and Pang-jo Chun

submitted to *The Cryosphere*

**Manuscript summary**

The manuscript describes the usage of machine learning (ML) models on glacier mass balance in Central Asia. The introduction of Temporal Fusion Transformers is investigated against more simple ML approaches with the aim of improving glacier mass balance predictions, as well as identifying the specific role of certain environmental conditions on the annual glacier mass balance. A freely available data set of annual glacier mass balances was used in combination with monthly meteorological information from ERA5 and glacier specific geographic information from RGI7. The mass balance data set spanning from 2000 to 2028 was split into a training subset (2000-2016) a validation subset (2017) and a prediction subset (2018). Based on the correlation statistics the quality of the different ML applications was evaluated for predicting future glacier mass balance, as well as providing information about the governing processes ("temporal importance patterns").

**General comments**

ML approaches are increasingly used in glaciology for different applications, from feature mapping and mass balance estimations to speeding-up ice dynamic glacier models. Especially if large samples need to be analysed, ML methods might be helpful to detect patterns and improve the process chain. Also for the prediction of glacier mass balance ML techniques have a strong potential to handle large data sets efficiently. This study aims to apply advanced ML techniques on such a large data set of glacier mass balance series in Central Asia. Unfortunately the study contains a series of flaws, wrong assumptions, unclear descriptions and obvious confusions. Because already the basic input data and the accompanied test setup are flawed, I suggest to reject this manuscript to allow a fundamental revision.

Barandun et al. (2021) used volume change time series from 1995 glaciers in the Tian Shan and Pamir mountains as calibration for their simple degree day model for reconstructing mass balance time series. However, they only used 1222 glacier time series in the end, due to constraints in their quality assessment, for a detailed analysis of spatial and temporal variability of glacier mass balances in this region. Provided are annual data sets for these glaciers for the period 2000-2018. This results in 23218 data values. Avzalshoev and Chun state that the use a total of 4 659 264 monthly samples from Barandun et al. (2021). I wonder which data they used.

Unfortunately, there is an obvious misunderstanding that the Barandun et al. (2021) data set consists of annual mass balance data derived from altimetry and photogrammetry, which is not the case. The data are the product of a simple model, calibrated by snow line data and geodetic volume differences. These data, therefore, are themselves reconstructed by using meteorological input from ERA5. Applying an ML routine to reconstruct a mass balance series, which is based on a simple degree day model and ERA5 input by using ERA5 data will provide no additional information at all, besides the ability of the ML routine to mimic a given degree day model. This approach is per se not suitable to investigate any parameters related to real glacier response to meteorological conditions.

Even though the manuscript looks rather well written at the first glimpse, it turns out that it contains a number of flaws at a closer look. Figures are not well described, while the captions are in many cases not related to the figures. E.g. there is no colour coding of the mountain ranges in Fig. 1, Fig. 2 shows no a), b), c) and d), Fig. 4 shows something completely different as the caption suggests, Fig. 5 should be a scatter plot according to the text, while the caption does again not fit to the data shown, Fig. 6-9 also show wrong captions. There are many more inconsistencies, which I can provide to the authors upon request. However, the manuscript needs to be completely rewritten in my opinion which renders it unnecessary to work on such details.

Regarding the scientific value of the manuscript Fig. 10 demonstrates that there likely exists a fundamental misunderstanding of the authors regarding the suitability of models o glaciological science. In line 353 ff. the authors state: "The annual comparison reveals excellent predictive performance with a mean accuracy of 92.6% for the test period (2017-2018).", which seems a purely model generated metric. The overestimation of the mass loss in Fig. 10 with regards to the simulated mass loss of Barandun et al. (2021) is about 60% which is a poor result and not an excellent predictive performance as stated by the authors. The application of the model to the prediction of one mass balance year only makes the entire discussion of the model performance, predictive processes etc. highly unreliable.

I stop here with my review, because the problems in this manuscript are so serious, that it requires a full reconsideration by the authors.